# Effectiveness of Silver Diammine Fluoride Applications for Dental Caries Cessation in Tribal Preschool Children in India: Study Protocol for a Randomized Controlled Trial

**DOI:** 10.3390/mps4020030

**Published:** 2021-05-11

**Authors:** Chandrashekar Janakiram, Venkitachalam Ramanarayanan, Induja Devan

**Affiliations:** Department of Public Health Dentistry, Amrita School of Dentistry, Amrita Vishwa Vidyapeetham, Kochi Kerala 682041, India; venkitr2006@gmail.com (V.R.); indujadevan@gmail.com (I.D.)

**Keywords:** Silver Diammine Fluoride, dental caries, disease progression

## Abstract

Introduction: Silver Diammine Fluoride (SDF) is an emerging caries preventive treatment option that is inexpensive, safe, and easily accessible. The evidence is clear that the use of SDF at concentrations of 38% is effective for arresting caries in primary teeth. However, the determination of an optimal SDF application frequency for a cavitated lesion in pragmatic settings is warranted especially among high dental caries risk groups. Hence, the primary objective of this clinical trial is to compare the effectiveness of annual, bi-annual, and four times a year application of 38% SDF application in arresting active coronal dentinal carious lesions on primary teeth among tribal preschool children aged 2–6 years. Methods and Analysis: This study is designed as a randomized, controlled trial consisting of three parallel arms with an allocation ratio of 1:1:1. The trial will enroll 480 preschool tribal children with a cavitated carious lesion (2–6 years) attending a primary health care Centre in Wayanad district, India. Each arm will receive 38% SDF application on an annual (baseline), bi-annual (baseline and 6 months), and four times a year (baseline, 2nd, 4th, and 8th week), respectively. The analysis will be performed both at the tooth- and person-level. Ethics and Dissemination: This trial will be conducted following the principles of the Declaration of Helsinki and local guidelines (Indian Council of Medical Research). The protocol has been approved by Institutional Review Committee (IRB). This trial has been registered prospectively with the Clinical Trial Registry of India [Registration No: CTRI/2020/03/024265].

## 1. Introduction

Dental caries is the most common chronic childhood disease, posing a significant public health problem [1]. Oral diseases have considerable social and economic impacts on the individual and community [2]. Early Childhood Caries (ECC) is a major public health problem worldwide. ECC prevalence varies, but the underprivileged children, across cultures, have a greater proportion of vulnerability [3]. The prevalence of ECC in Indian preschool children ranges from 12% to 94.3% [4,5,6,7]. Untreated ECC can cause pain and can lead to multiple consequences such as difficulty in eating, speaking, and attending to learning as it occurs during the years of milestone development of the child [8]. 

ECC are relatively inexpensive to prevent [9,10,11,12,13], yet becomes extremely burdensome on the children and families, and expensive to treat once lesions cavitate in young children who need extensive treatment or are uncooperative and/or have immature cognitive functioning, disabilities, or medical conditions, where treatment under general anesthesia, in most cases in hospital operating rooms, is the standard of care. Effective ECC preventive measures include the use of fluoride varnishes such as 5% Sodium Fluoride (NaF) and the use of fluoridated toothpaste [14]. These agents are beneficial before the onset of ECC. However, if caries has progressed to cavitation due to failure of preventive strategies, then there is a need to arrest the progression of the condition. In the management of cavitated ECC, removal of infected tooth tissue and restoration is recommended which is resource intensive as discussed earlier.

The 38% SDF, approved as a desensitizing agent, has been used (off-label) as a caries-arresting agent. SDF is a water-like liquid that is applied to teeth with active carious lesions. It has a dual effect against caries owing to the presence of fluoride which promotes remineralization and silver which provides antimicrobial action. Furthermore, when bacteria killed by the silver ions are added to the living bacteria, the silver is reactivated thereby effectively killing the live bacteria eliciting a zombie effect [15]. A highly re-mineralized layer with high calcium and phosphate content was observed on arrested dentine caries lesions. Thus, SDF as a chemotherapeutic agent can be used in cavitated lesions of ECC to stop the progression of dental caries.

There have been multiple clinical trials and several systematic reviews reporting the application of SDF causing the arrest of caries lesions in a high percentage of cases (30–70%) [16,17,18,19,20,21,22,23,24]. Nearly 70% of caries lesions in primary teeth would be expected to be arrested in two years after SDF application with the variation of annual or biannual application as compared to combined controls of 5% NaF varnish or restorations. According to the AAPD manual [25], nearly 68% (95% CI = 9.7 to 97.7) of cavitated caries lesions in primary teeth would be expected to be arrested in two years after SDF application with a variety of annual or biannual application as compared combined controls of 5% Sodium fluoride (NAF) varnish or restorations. The American Dental Association (ADA) guidelines recommend the use of 38% SDF twice a year application over 5% NaF varnish (once per week for 3 weeks application) to arrest the cavitated caries coronal lesions in primary and permanent teeth [16]. However, it is a conditional recommendation that implies that not all patients would benefit from the intervention and depends on individual risk for caries, preferences, values, etc.

Studies are comparing various frequencies (thrice a year, weekly, biannual, annual, five times weekly per year, etc.) of SDF; however, the evidence is inconsistent. A clinical trial has shown that the yearly application of SDF has a higher proportion of arrested caries compared to the weekly application [26]. Biannual application of SDF was found to be superior compared to annual application in arresting cavitated carious lesion [27,28,29,30,31]. Annual application of silver diamine fluoride prevented many more carious lesions than application of fluoride varnish four times per year [26,31].

Tribal children face many health disparities compared with the mainland counterparts [32]. The prevalence of ECC is 74.7% in these tribal children compared to their counterparts [7]. The oral health disparities for these children exemplify, many of the inequities and early care services in these communities. SDF as a chemotherapeutic agent for prevention and managing of ECC would be a crunch point as it is a simple technique, acceptable, and cost-effective strategy over restorations and hospital-based management. The SDF effectiveness in these high-risk caries children will generate the evidence to use SDF as a caries preventive tool to the population having less access to regular dental care. The trial findings may also help in reducing pain, improving quality of life, and significantly reducing costs, all contributing to substantial reductions in disparities in caries the disparities in between access to preventive care. The use of SDF to prevent or delay surgical intervention until after the age of 3 years makes it a potentially attractive adjunctive therapy for managing caries in the young pediatric population.

Though the evidence is clear that the use of 38% SDF is effective for arresting dental caries, the determination of an optimal SDF application frequency for a cavitated lesion in pragmatic settings is important especially among high caries risk tribal children who have poor access to oral care. Thus, the primary objective of this trial is to compare the effectiveness of an annual, semi-annual, and four times a year application of 38% Silver Diammine Fluoride in arresting active coronal carious lesions on primary teeth of tribal preschool children in India. The secondary objective is to assess the acceptability of SDF treatment by the child as well as parent/s.

## 2. Methods

### 2.1. Study Design

This trial is designed as a randomized, controlled trial consisting of three parallel arms with an allocation ratio of 1:1:1 (Figure 1).

### 2.2. Study Settings

The trial will be conducted in a primary health care center at Kalpetta, Kerala, India. Kalpetta is the headquarters of Wayanad district which has the largest tribal population in the state of Kerala.

### 2.3. Trial Registration

The trial protocol has been registered with the Clinical Trial Registry of India at http://ctri.nic.in accessed on 10 May 2021. (Registration No CTRI/2020/03/024265 dated Registered on 25 March 2020). Recruitment of study subjects will begin in April 2021 and the trial is expected to conclude by December 2023. This clinical trial protocol has been developed following the SPIRIT 2013 Statement (https://www.spirit-statement.org/ accessed on 10 May 2021).

### 2.4. Study Population

Tribal preschool children aged two to six years attending the primary health care center for routine dental/medical care and with one or more cavitated dentinal carious lesion present in any surface of the primary teeth will constitute the sampling frame for the study.

### 2.5. Eligibility Criteria

#### 2.5.1. Inclusion Criteria

Children of any gender aged 2–6 years with the presence of cavitated caries lesion classified as stage 5–6 following the Modified International Caries Detection and Assessment System (ICDAS).The parent of the child understands the importance of SDF treatment to arrest prevent dental caries.Parent of a child willing to accompany the child at required intervals for evaluation.

#### 2.5.2. Exclusion Criteria

Children with ulcerative gingivitis or stomatitis.Developmental dental abnormalities such as enamel defects.Serious chronic medical conditions, such as congenital heart disease.Cavitated teeth that are nearing natural exfoliation time.Children with more than one-third of the crown missing, or pulpal involved (tooth with pulp exposure, presence of an abscess or a sinus, obvious discoloration, and premature hypermobility were regarded as pulpal involved tooth)Known silver allergy.Uncooperative children (after the failure of basic pediatric patient management protocols)Children with special health care needs.

### 2.6. Definition of Study Condition (Dental Caries)

A cavitated lesion is a carious lesion with a surface that is not macroscopically intact and with a distinct discontinuity or breaks in the surface integrity [33]. The tooth should be classified as stage 5 or 6 according to Modified International Caries Detection and Assessment System (ICDAS) [25].

The status of the dentinal caries lesions will be assessed by visual inspection aided by tactile detection using a Community Periodontal Index (CPI) probe. An artificial light source from the dental chair will be used. The tooth surface will be dried and a CPI probe with the tip gently passed over the entire surface of the cavity to detect and confirm the visual evidence of caries. Cavities with yellowish/brown rough wall/floor which could be easily penetrated by the probe will be diagnosed as active. Five surfaces in each posterior tooth (occlusal, buccal, lingual, mesial, and distal) and four surfaces in each anterior tooth (labial, lingual, mesial, and distal) will be assessed. Radiographic aids will not be used for the diagnosis of caries.

### 2.7. Interventions

#### Silver Diammine Fluoride

SDF is a colorless liquid containing silver ionic particles. It contains 38% (44,800 ppm) fluoride ion at a pH of 10. This formulation constitutes 25% silver, 8% ammonia, 5% fluoride, and 62% water. One drop (0.05 mL) of 38% SDF solution (2.24 F-ion mg/dose) is required for the treatment of one tooth. The SDF treatment protocol consists of removing gross debris from the cavitated tooth, drying the affected area, isolating the tooth with the help of cotton rolls and suction, placing a small amount of SDF on the affected area, gently drying the SDF for 1 min, removing excess SDF with cotton gauze and isolating the tooth for up to 3 min [34]. This procedure will be performed after the tooth is removed of debris (with the aid of a toothbrush) and soft caries using a spoon excavator. It includes gross debris from cavitation to allow better SDF contact with denatured dentin. Carious dentin excavation before SDF application is not necessary. As excavation may reduce the proportion of arrested caries lesions that become black, it may be considered for esthetic purposes. No rotary instruments will be used for caries removal.

The study participants will be allocated into one of the three parallel arms who will receive 38% SDF application as follows:
Study arm 1: annually (once a year; baseline visit);Study arm 2: bi-annually (twice a year at 6-month intervals; baseline visit and after 6 months);Study arm 3: four times a year (baseline visit, 2nd, 4th, and 8th week).

In the baseline visit in all the study arms, the child will be assessed for cavitated caries lesions and caries risk (ADA Caries Risk Assessment Form (Age 0–6)). After obtaining consent for the trial, SDF is applied on the selected caries teeth depending on the study arm. The demographic characteristics, plaque index, and caries status will be assessed. In the subsequent visits depending on the study arm, SDF will be applied (Table 1).

### 2.8. Primary Outcomes

The primary outcome of the study is the arrest of caries. The hardness of treated cavitated surfaces of teeth on probing (periodontal probe) is an indication that a lesion is arrested. Blackish discoloration of surface/cavity alone will not be a criterion for defining the arrested caries.

The outcome assessment will be performed for all the arms at 6 months and 12 months. SDF treated cavities of teeth with arrested caries are defined as surfaces with active cavitated caries at baseline that changed into surfaces with arrested cavitated caries on subsequent assessments. The possible scenarios and their assessment are as follows:Any teeth with treated carious lesions that are lost due to exfoliation will be considered arrested throughout the lifetime of the tooth. For any reason, if the child’s teeth are restored before the trial period, censoring of the data depends on the duration of the trial period the child has participated.The following dental complications due to the evolution of caries will not be considered as caries arrested.○The teeth extracted due to caries progression.○If the SDF treated cavities, caries progressed beyond the dentine.○If the tooth is extracted due to fracture, trauma will not be counted for the study.

### 2.9. Secondary Outcome

To assess the children (and/or parents) satisfaction and acceptability and dentist factors associated with the use of the SDF. This will be assessed with the help of a self-reported questionnaire.

### 2.10. Participation Timeline

Each eligible child will be a part of the trial for one year from their baseline visit. The total number of visits depends on the study arms to which child is allocated (Figure 1).

### 2.11. Sample Size

The sample size was calculated based on the primary objective. In an exploration study [26], those children treated weekly with 38% SDF had a slightly higher proportion of arrested caries compared to annual application. There was no considerable difference in the proportion of caries arresting by either semiannual or annual application (less than 10%). We expect that the proportion of the arrested carious lesion not to be more than 14% between the two frequencies of application. Considering a slightly higher proportion of caries arrest by the four times application as compared to others, we estimated the sample size at the power of 0.80 and alpha error of 0.05 to be 120 per arm. Being a population (tribal), which is not mainstream and owing to the demands of multiple follows of the tribal population, it is expected that we could encounter a lost-to-follow-up of about 30%. Accordingly, the final sample size has been increased to 160 children per arm. In total, 480 children will be enrolled on the trial.

### 2.12. Study Implementation

The 2–6-year-old children visiting the primary health care center (Amrita Kripa Health Care Centre) for routine dental/medical care or accompanying parents requiring dental/medical care will be invited to be a part of the study. On dental examination, if the child is found to satisfy the inclusion criteria, then the parent of the child will be given the option for treating the carious lesion by SDF treatment or by routine restoration. If the parent opts for SDF treatment for the child, then an option to participate in the trial will be given. If the parent agrees, then the consenting process is followed. Informed consent in the local language will be obtained from willing participants.

The selected child for trial will be randomly allocated to one of the three intervention arms as per computer-generated randomization sequence stratified at 40% type of tooth (anterior) and 50% gender. In this process, stratification is performed by gender (males, females) and type of primary tooth (anterior, posterior), so will have a total of 4 strata (2 by 2). Treatment assignments are then made from separate computer-generated randomization lists created in advance for each stratum. This randomization sequence will be generated from a remote site independent from the knowledge of the Principal Investigator (PI).

The randomization will be completed by a trial coordinator based on the computer-generated randomization sequence. The randomization list will remain with the trial coordinator until the completion of the study. The intervention (SDF application) will be performed by the PI and outcome assessment will be performed by two investigators other than the PI. Due to the obvious nature of the treatment neither the investigator performing the treatment, nor the study participant can be blinded. However, the outcome assessors are proposed to be blinded.

Allocation concealment will be conducted using opaque sealed envelopes (sealed sequentially numbered envelops irreversibly) containing codes (corresponding to the randomization list). The allocation concealment will be performed by one author (RV). The allocation sequence will not be disclosed to the PI until the completion of the study.

The study requirements, ensuring standardization in the assessment of the cavitated dental lesion, assessment of arrested lesion, application of the 38% SDF varnish to the child, counselling for adherence and eliciting of information from study participants in a uniform reproducible manner will be undertaken before the beginning of the trial at Amrita Kripa Health Care Centre. The data to be collected and the procedures to be conducted at each visit will be reinforced to all the personnel related to the trial. Entering data forms, responding to data discrepancy queries and general information about obtaining research quality data will also be covered during the training session. Calibration will be performed before and during the trial, and reliability expressed using Cohen’s kappa statistic. Once a child is enrolled or randomized, the study site (Amrita Kripa Health Care Centre) will make every reasonable effort to follow the child for the entire study period of 12 months.

### 2.13. Compliance with Trial Treatment

Compliance to visit the dental clinic at designated intervals will be conducted by text messages/telephonic reminders to parents. If the child misses an appointment or wants a reschedule, then it is rescheduled with a plus or minus of three days. If the child misses two rescheduled appointment, then it is referred to as non-compliant and the child is withdrawn from the study. However, if the child completes six months (1st outcome assessment) and then becomes non-compliant, the person-time of the participation in the trial will be calculated and will be used for analysis.

### 2.14. Adverse Events

The 38% SDF used in this trial is employed in routine clinical practice as a standard of care procedure. The potential risks for a child are minimal and identical to the risk for children obtaining routine dental care. The only foreseeable risk would be an allergic reaction to SDF. Silver diamine fluoride stains clinic surfaces and clothes. At each contact with the study participant, investigators will seek information on adverse events by specific questions and an oral examination.

The child may experience a transient metallic or bitter taste. Even a small amount of silver diammine fluoride can cause a “temporary tattoo” to the skin (on the patient or dentist). The patient will be informed about the same during the consenting process and reinforced during the follow-up visits. If the child/parent refuses to participate at any point in time owing to any of the above-mentioned conditions, he/she will be considered as a drop-out. Stain on the skin resolves with the natural exfoliation of the skin, usually in 2–14 days. Universal precautions will prevent most exposures to the adjacent tissues. Isolation of the areas to be treated with cotton rolls and application cocoa butter to protect surrounding gingival tissues will prevent inadvertent exposure. Spills can be cleaned up immediately with copious water, ethanol, or bleach. High pH solvents such as ammonia may also be used. Secondary containers and plastic liners for surfaces are adequate preventives.

## 3. Close-Out Procedures

The end of the trial is the date of the last visit of the last participant randomized which is 12 months from the baseline. Close-out will proceed in two stages: an interim period for analysis and documentation of study results and the second one would be a debriefing of participants and dissemination of study results.

## 4. Statistical Methods

The outcome of the treated carious lesion is binary, either arrested lesion or failed to arrest at 6 months follow-up and at the end of 12 months follow up visit. The outcome will be assessed at two levels: Tooth-level—the proportion of cavitated surfaces at baseline will be compared with the proportion of the surfaces arrested at each frequency of application at 6 months follow-up and the end 12 months using chi-square methods. Between three frequencies of application (annual, semi-annual, and four times a year applications) the proportion of arrested surface at 6 months follow-up and end 12 months will be compared. This comparison will be stratified by type of tooth (posterior vs. anterior) and type of surfaces (facials vs. lingual). Child-level—each child may have one or more cavitated carious surface treated with SDF. So, the per-child proportion of cavitated carious lesions at baseline treated with 38% SDF applied at different frequencies that stayed arrested throughout observation will be determined.

The regression will be modelled on the dependent variable as a percentage of the arrested carious lesion per child with another. The two sets of analysis will be done, first, the intention-to-treat set, considering all patients as randomized regardless of whether they received the randomized treatment, and second, the “per protocol” analysis set. The hypothesis that four times application of 38% of SDF superior to other frequencies (annual or semi-annual application) is accepted, only if shown to be superior using both the “intention to treat” and “per protocol” analysis sets.

## 5. Analysis Population and Missing Data

The reasons for withdrawal in each group will be assessed and reported qualitatively. The effect that any missing data might have on results will be assessed via sensitivity analysis of augmented data sets. Dropouts (after six months) will be included in the analysis and values will be imputed for missing data. If the attrition loss is less than 15%, “per protocol analysis” will be carried out otherwise “Intention to treat” will be performed.

## 6. Patient and Public Involvement

The tribal communities are geographically isolated and extremely deprived. They face huge health disparities with the mainland population. Coping with the burden of more life-threatening diseases, oral health is seen as less of a priority. As a result, dental disease, and the lack of awareness about oral health, result in the extraction of teeth and thereafter a compromised life due to edentulism. They face affordability, accessibility, and availability of health care issues. Dental caries among children and tobacco-related oral diseases are common among adults. The research group and institution of the study have initiated the outreach program for dental care for these tribal population for the past 12 years (https://www.amrita.edu/school/dentistry/amritasmitham accessed on 10 May 2021). The research idea and design of the study was developed based on the experiences gained from the program. SDF would be ideal for cessation of dental care in these high caries risk tribal children. SDF is a chemotherapeutic, non-surgical treatment with SDF that requires minimal instrumentation and application at less frequent intervals than other caries preventive materials [15]. Current evidence supports SDF to use in children and can be potentially adjunctive therapy for preventing or delay surgical intervention for managing caries in young children. However, the effective frequency of the application seems too less known.

Local tribal community leaders will be involved in commenting and developing patient information leaflets or other research materials and undertaking interviews with the participants. We will be selecting research assistants from the local community to help in the recruitment and follow-up process of the study. Moreover, awareness regarding the importance of oral health and good oral hygiene is being planned for the tribal community. The study findings are proposed to be disseminated to research participants, tribal community leaders, and local health authorities.

## 7. Ethics and Dissemination

This trial will be conducted by the principles of the Declaration of Helsinki and local guidelines (Indian Council of Medical Research). The protocol has been approved by the Institutional Review Board (IRB) of Amrita Institute of Medical Sciences (AIMS), Kochi, India (IRB-AIMS-2019-287). The investigator shall submit once a year throughout the clinical trial, or on request, an Annual Progress Report to the IRB, AIMS. Additionally, an End of Trial notification and a final report will be submitted. No unauthorized persons will have access to final data sets.

A trained research assistant will introduce the trial to patients the main aspects of the SDF trial. Patients will also receive printed information sheets. Patients will then be allowed to have an informed discussion. Parent/s of the child will be detailed no less than the exact nature of the trial; what it will involve for the participant; the implications and constraints of the protocol; the known side effects and any risks involved in taking part in the trial. It will be clearly stated that the participant is free to withdraw from the trial at any time for any reason without prejudice to future care, without affecting their legal rights and with no obligation to give the reason for withdrawal. Information sheets and consent forms will be provided for all parents involved in the trial. All information sheets, consent forms will be in study subjects speaking the language. All presentations and publications are expected to protect the integrity of the major objective(s) of the study; data that break the blind will not be presented before the release of mainline results. No later than 3 years after the collection of the 1-year post-randomization, the completely de-identified data set will be shared with an appropriate data archive source. The study results and actionable recommendations proposed will be shared with the local health authorities and community leaders.

## Figures and Tables

**Figure 1 mps-04-00030-f001:**
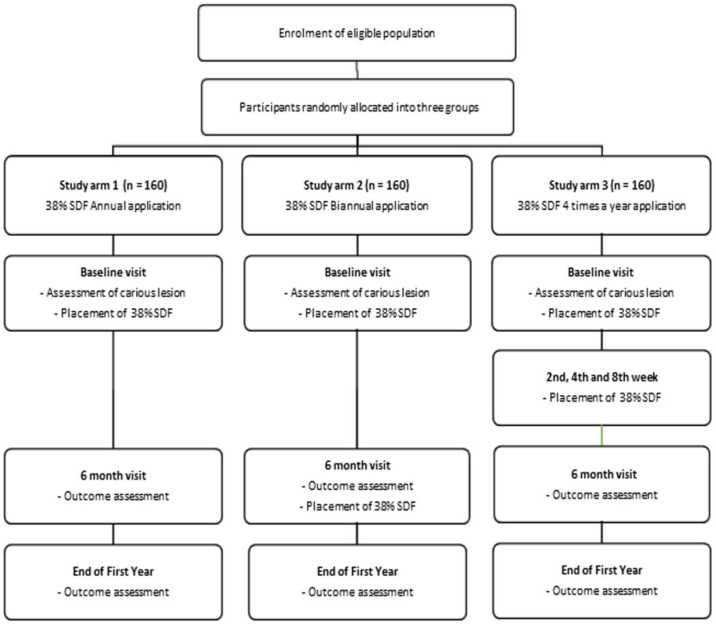
Study flow diagram.

**Table 1 mps-04-00030-t001:** Participation timeline of the trial.

Procedures		Timeline					
		Visit 1	2	3	4	5	6
		Baseline	2 Weeks	4 Weeks	8 Weeks	6 Months	12 Months
Pre-screening consent		Yes	-	-	-	-	-
Eligibility assessment		Yes	-	-	-	-	-
Oral examination		Yes	Yes	Yes	Yes	Yes	Yes
Informed consent		Yes	-	-	-	-	-
Allocation to study arms		Yes	-	-	-	-	-
Caries Excavation		Yes					
Application of 38% SDF	Group 1	Yes	No	No	No	No	No
	Group 2	Yes	No	No	No	Yes	No
	Group 3	Yes	Yes	Yes	Yes	No	No
Outcome assessment		Yes *	-	-	-	Yes	Yes
Adverse event assessments		Yes	Yes	Yes	Yes	Yes	Yes

* Baseline assessment.

## Data Availability

Data sharing is not applicable to this article.

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
