# Peer review of "Effectiveness of Silver Diammine Fluoride Applications for Dental Caries Cessation in Tribal Preschool Children in India: Study Protocol for a Randomized Controlled Trial"

_mps, 2021, doi:10.3390/mps4020030_

Round 1

Reviewer 1 Report

check the font of the text
review the numbering of bibliographic citations
I suggest inserting a space before the list of exclusion rules.

Author Response

Thanks for the review and suggestions. 

Query 1. check the font of the text

Reply : The font size (11) of the manuscript has been formatted uniformly. 

Query 2. Review the numbering of bibliographic citations

Reply:  The numbering of citations has been corrected. 

Query 3: I suggest inserting a space before the list of exclusion rules.

Reply: The typo error has been corrected. 

Reviewer 2 Report

The referee agrees with the publication as it is.

Author Response

Comments: The referee agrees with the publication as it is.

Reply Thank you for your review.

Reviewer 3 Report

High quality protocol. Just one point. You write for the sample size calculation: 
In total, 480 children will be enrolled for the trial. You do not take into account a loss of 10-20% of subjects, which may limit the power of your results?

Author Response

Comments: High-quality protocol. Just one point. You write for the sample size calculation: 
In total, 480 children will be enrolled for  the trial. You do not take into account a loss of 10-20% of subjects, which may limit the power of your results?

The detailed sample size estimation has been described in the Methods section of the manuscript (line 235 onwards)  and anticipated attrition loss has been adjusted to the total sample size 

Sample size

The sample size was calculated based on the primary objective. In an exploration study,[38] those children treated weekly with 38% SDF had a slightly higher proportion of arrested caries compared to annual application. There was no considerable difference in the proportion of caries arresting by either semiannual or annual application (less than 10%). We expect that the proportion of the arrested carious lesion not to be more than 14% between the two frequencies of application. Considering a slightly higher proportion of caries arrest by the four times application as compared to others, we estimated the sample size at the power of 0.80 and alpha error of 0.05 to be 120 per arm. Being a population (tribal), which is not mainstream and owing to the demands of multiple follows of the tribal population, it is expected that we could encounter a lost-to-follow-up of about 30%. Accordingly, the final sample size has been increased to 160 children per arm. In total, 480 children will be enrolled on the trial.

Reviewer 4 Report

1. 
Spelling - the technically correct spelling is diammine - not diamine (based on chemical nomenclature).  Define SDF as an abbreviation and use that throughout.

2. For stylistic consistency,

the effectiveness of annual, bi-annual and 4 times a year application

should be

the effectiveness of annual, bi-annual and quarterly application

3.
Hence, the primary objective of this clinical trial  (missing word "the")

4.
The secondary objective is to assess the acceptability 
(missing word "the")

5. The introduction has font size inconsistencies

6.
In the introduction a rather sweeping statement is made: "Several evidence-based approaches for caries prevention have been recommended; however, these strategies are resource intensive, that are costly to any community.[9]" This statement does not cover a broad range of interventions for ECC that have been shown to be highly cost effective, and NOT resource intensive, such as those described in the following papers:
PMID: 31470927 PMID: 23674443 PMID: 26110399 
PMID: 23488214 PMID: 14696738
Hence, this sentence must be reworded.

7.
Later in the introduction the following sweeping statement is made " ART is .... unlikely to permanently stop progression of caries due to the possibility of the presence of infected dentin". This is NOT correct - either for ART or for traditional stepwise caries removal. There are many papers on this topic over 20 years. I suggest look at the considerable literature on ART clinical outcomes,

The introduction would much better be written to purely focus on the key issue - which is - SDF works, but how often does it need to be applied to get caries arrest in socially disadvatnaged groups?

Stick to that topic and don't wander off and make broad and incorrect generalizations. Hence, comment on the range of application frequencies that have been used in clinical studies; also comment on the recent US and other recommended protocols for SDF use in ECC and how the situation could be different in communities with social disadvantage, e.g. because they lack basic oral hygiene devices like toothbrushes and fluoride toothpaste.

8.
The following paragraph has a major error. "A clinical trial has shown that yearly application of SDF has a higher proportion of arrested caries compared to weekly application. Biannual application of SDF was found to be superior by 15% compared to annual application in arresting cavitated carious lesion.[23]"  This is NOT what that paper actually shows, rather in that study "Group 1 - annual application of 30% SDF solution; Group 2 - three applications of 30% SDF at weekly intervals; and Group 3 - three applications of 5% NaF varnish at weekly intervals." Please correctly describe the past study.

9. The criterion "Parent of child understands importance of SDF treatment to prevent dental caries" should be "Parent of child understands importance of SDF treatment to arrest dental caries". This is NOT a study of caries prevention but a study where lesions are treated and arrest is the desired outcome.

10. "SDF is a colorless liquid containing silver particles." It contains ionic silver, not silver particles.  

11. There is no discussion of gingival tissue protection - SDF is well known to cause soft tissue burns of mild to moderate severity - yet this aspect is overlooked all together.

12. "soft caries using a spoon excavator" Insufficient information has been given. Is this step excavation, like ART or something else? What are the endpoints? Will the teeth in one group undergo multiple episodes of excavation and SDF? What about the risk of iatrogenic pulp exposure? Or patient discomfort. There is no explanation as to why excavation is being done - versus just using SDF alone.

13. What is the minimal number of lesions per child subject? Just 1? How will compliance be affected if multiple lesions are treated in the same subject?

14. Table 1 is missing a column for "Caries excavation"

15. The clinical outcome is unclear - why even mention a ball burnisher? 

16. How will examiners be calibrated in terms of probing force and technique? 

17. No description of adverse event recording is given.

18. Details of the q'aire are missing - e.g. will it look at soft tissue discomfort, tooth discomfort initially and over time, changes in tooth colour over time, etc.

19. The paper needs comprehensive English language editing - all sections have textual errors.

20. The references have numerous errors in style in the use of capitals in the title of articles 3, 7, 10, 18, 21, 26, 27.

Author Response

Dear Reviewer,

Thanks for reviewing the manuscript. We have made a substantial revision of the manuscript for your comments and suggestions have been incorporated. The changes are highlighted in the red color text in the manuscript attached. 

1.Spelling - the technically correct spelling is diammine - not diamine (based on chemical nomenclature).  Define SDF as an abbreviation and use that throughout.

Reply Thanks for noticing the typo error. Diammine will be used.  

  1. For stylistic consistency, the effectiveness of annual, bi-annual and 4 times a year application should be typo the effectiveness of annual, bi-annual and quarterly application

Reply:  The term quarterly is used usually when the application is once in 3 months. But here it is applied at 0, 2, 4, and 8 weeks. We feel the use of the word quarterly may be misinterpreted by readers and hence we choose to retain the word four times a year. 

3.Hence, the primary objective of this clinical trial (the missing word "the")

Reply: The typographical error of the objective has been corrected

4.The secondary objective is to assess the acceptability (the missing word "the")

Reply: The typographical error of the objective has been corrected

  1. The introduction has font size inconsistencies

Reply:  Font size 11 is used throughout the manuscript 

  1. In the introduction a rather sweeping statement is made: "Several evidence-based approaches for caries prevention have been recommended; however, these strategies are resource intensive, that are costly to any community.[9]" This statement does not cover a broad range of interventions for ECC that have been shown to be highly cost effective, and NOT resource intensive, such as those described in the following papers:
    PMID: 31470927 PMID: 23674443 PMID: 26110399 
    PMID: 23488214 PMID: 14696738
    Hence, this sentence must be reworded.

Answer

Reply: The sentence has been changed accordingly to references to provided.

ECC is relatively inexpensive to prevent [9–13], yet becomes extremely burdensome on the children and families, and expensive to treat once lesions cavitate in young children who need extensive treatment or are uncooperative and/or have immature cognitive functioning, disabilities, or medical conditions, where treatment under general anaesthesia, in most cases in hospital operating rooms, is the standard of care. 

  1. Later in the introduction, the following sweeping statement is made " ART is .... unlikely to permanently stop the progression of caries due to the possibility of the presence of infected dentin". This is NOT correct - either for ART or for traditional stepwise caries removal. There are many papers on this topic over 20 years. I suggest looking at the considerable literature on ART clinical outcomes,

The introduction would much better be written to purely focus on the key issue - which is - SDF works, but how often does it need to be applied to get caries arrest in socially disadvantaged groups?

Stick to that topic and don't wander off and make broad and incorrect generalizations. Hence, comment on the range of application frequencies that have been used in clinical studies; also comment on the recent US and other recommended protocols for SDF use in ECC and how the situation could be different in communities with social disadvantage, e.g. because they lack basic oral hygiene devices like toothbrushes and fluoride toothpaste.

Reply: Thanks for the suggestion to change the narration. We have removed a discussion on ART and followed up with your suggestions. The introduction has been revised.

8.
The following paragraph has a major error. "A clinical trial has shown that yearly application of SDF has a higher proportion of arrested caries compared to weekly application. Biannual application of SDF was found to be superior by 15% compared to annual application in arresting cavitated carious lesion.[23]"  This is NOT what that paper actually shows, rather in that study "Group 1 - annual application of 30% SDF solution; Group 2 - three applications of 30% SDF at weekly intervals; and Group 3 - three applications of 5% NaF varnish at weekly intervals." Please correctly describe the past study.

Reply Thanks for noticing this error. We have corrected the description of the study properly and added more studies.

  1. The criterion "Parent of child understands importance of SDF treatment to prevent dental caries" should be "Parent of child understands importance of SDF treatment to arrest dental caries". This is NOT a study of caries prevention but a study where lesions are treated and arrest is the desired outcome.

Reply: The sentence is corrected.

  1. "SDF is a colourless liquid containing silver particles." It contains ionic silver, not silver particles.  

Reply: The sentence is corrected.

  1. There is no discussion of gingival tissue protection - SDF is well known to cause soft tissue burns of mild to moderate severity - yet this aspect is overlooked all together.

Reply: We have mentioned in the exclusion criteria, that “children with ulcerative gingivitis or stomatitis excluded. In the case of gingiva tissue irritation, there is a description.

Universal precautions will prevent most exposures to the adjacent tissues. Isolation of the areas to be treated with cotton rolls and application cocoa butter to protect surrounding gingival tissues will prevent inadvertent exposure. 

  1. "soft caries using a spoon excavator" Insufficient information has been given. Is this step excavation, like ART or something else? What are the endpoints? Will the teeth in one group undergo multiple episodes of excavation and SDF? What about the risk of iatrogenic pulp exposure? Or patient discomfort. There is no explanation as to why excavation is being done - versus just using SDF alone.

It includes gross debris from cavitation to allow better SDF contact with denatured dentin. Carious dentin excavation before SDF application is not necessary. As excavation may reduce the proportion of arrested caries lesions that become black, it may be considered for aesthetic purposes.” 

  1. What is the minimal number of lesions per child subject? Just 1? How will compliance be affected if multiple lesions are treated in the same subject?

The minimum number of lesions per child is one. Multiple lesions in the same child will receive the same intervention group. Hence compliance is not an issue. The compliance measures to be taken have been described in the methods section.

  1. Table 1 is missing a column for "Caries excavation"

Caries excavation is added in table 1

  1. The clinical outcome is unclear - why even mention a ball burnisher? 

Clinical outcome is the arrest of caries assessed by tactile examination, by a periodontal probe which has clearly described in the Primary outcomes. Ball burnisher was mentioned since it has a rounded end.

  1. How will examiners be calibrated in terms of probing force and technique? 

The study requirements, ensuring standardization in the assessment of the cavitated dental lesion, assessment of arrested lesion, application of the 38% SDF varnish to the child, counseling for adherence, and eliciting of information from study participants in a uniform reproducible manner will be undertaken before the beginning of the trial at Amrita Kripa Health Care Centre. The data to be collected and the procedures to be conducted at each visit will be reinforced to all the personnel related to the trial. Entering data forms, responding to data discrepancy queries and general information about obtaining research quality data will also be covered during the training session. Calibration will be done before & during the trial, and reliability expressed using Cohen’s kappa statistic. Once a child is enrolled or randomized, the study site (Amrita Kripa health care center) will make every reasonable effort to follow the child for the entire study period of 12 months.

  1. No description of adverse event recording is given.

A detailed description of the reporting of the adverse event has reported in the Methods section

Adverse events

The 38% SDF used in this trial is employed in routine clinical practice as a standard of care procedure. The potential risks for the child are minimal and identical to the risk for children obtaining routine dental care. The only foreseeable risk would be an allergic reaction to SDF. Silver diamine fluoride stains clinic surfaces and clothes. At each contact with the study participant, investigators will seek information on adverse events by specific questions and an oral examination.

The child may experience a transient metallic or bitter taste.  Even a small amount of silver diammine fluoride can cause a “temporary tattoo” to the skin (on the patient or dentist). The patient will be informed about the same during the consenting process and reinforced during the follow-up visits. If the child/parent refuses to participate at any point in time owing to any of the above-mentioned conditions, he/she will be considered as a drop-out. Stain on the skin resolves with the natural exfoliation of the skin, usually in 2-14 days. Universal precautions will prevent most exposures to the adjacent tissues. Isolation of the areas to be treated with cotton rolls and application cocoa butter to protect surrounding gingival tissues will prevent inadvertent exposure.  Silver diammine fluoride also stains clinic surfaces and clothes. Spills can be cleaned up immediately with copious water, ethanol, or bleach. High pH solvents such as ammonia may also be used. Secondary containers and plastic liners for surfaces are adequate preventives.

  1. Details of the q'aire are missing - e.g. will it look at soft tissue discomfort, tooth discomfort initially and over time, changes in tooth colour over time, etc.

Reply

The soft tissue discomfort will be assessed at each visit and all adverse events will be reported which has been described in adverse event sections.

A questionnaire will be given to the parent/child after each visit to assess self-reported oral hygiene performance and acceptability of the treatment.

Acceptability of the SDF treatments by the children and the treating dentist will be binary (preferred/not preferred to the restorations).

Acceptability to the child and their parent and ease of use will be measured by

The practicality of use on the children using surveys of parents and the dentist.

Compliance of children to treatment (Proportion of children who accept treatment).

  1. The paper needs comprehensive English language editing - all sections have textual errors.

We have made a substantial revision of the manuscript with professional language editors.

  1. The references have numerous errors in style in the use of capitals in the title of articles 3, 7, 10, 18, 21, 26, 27.

Errors in the reference have been corrected.

Round 2

Reviewer 4 Report

I am now broadly happy with this paper and the revisions address my earlier points.

The remaining task is to fix the references.

Despite my explicit comments earlier, namely "The references have numerous errors in style in the use of capitals in the title of articles"  the authors have still NOT fixed the style errors in the references,

e.g. for ref 7 the title should be  "Association of undernutrition and early childhood dental caries" and the same problem style occurs in refs 14, 25, 27, 29, 30, 32, 36 and 37. Remove the capitals to ensure a consistent style.

Ref 25 where there should be capitals in the title for the ICDAS words is missing them.

Ref 37 is also incomplete. 

These all need to be corrected.

Author Response

Dear Reviewer,

Thanks for noticing the referencing errors. The reference style has been revised.  There were two duplicate references that have also been corrected.

1             Nunn JH. The burden of oral ill health for children. Arch Dis Child 2006;91:251–3. doi:10.1136/adc.2005.077016

2             Sheiham A, Williams DM, Weyant RJ, et al. Billions with oral disease: A global health crisis--a call to action. J Am Dent Assoc 1939 2015;146:861–4. doi:10.1016/j.adaj.2015.09.019

3             Anil S, Anand PS. Early Childhood Caries: Prevalence, risk factors, and prevention. Front Pediatr 2017;(5):157. doi:10.3389/fped.2017.00157

4             Jose B, King NM. Early childhood caries lesions in preschool children in Kerala, India. Pediatr Dent 2003;25:594–600.

5             Priyadarshini HR, Hiremath SS, Puranik M, et al. Prevalence of early childhood caries among preschool children of low socioeconomic status in Bangalore city, India. J Int Soc Prev Community Dent 2011;1:27–30. doi:10.4103/2231-0762.86384

6             Prakash P, Subramaniam P, Durgesh BH, et al. Prevalence of early childhood caries and associated risk factors in preschool children of urban Bangalore, India: A cross-sectional study. Eur J Dent 2012;6:141–52.

7             Janakiram C, Antony B, Joseph J. Association of undernutrition and early childhood dental caries. Indian Pediatr 2018;55:683–5. doi:10.1007/s13312-018-1359-4

8             Çolak H, Dülgergil ÇT, Dalli M, et al. Early childhood caries update: A review of causes, diagnoses, and treatments. J Nat Sci Biol Med 2013;4:29–38. doi:10.4103/0976-9668.107257

9             Davies GM, Worthington HV, Ellwood RP, et al. An assessment of the cost effectiveness of a postal toothpaste programme to prevent caries among five-year-old children in the North West of England. Community Dent Health 2003;20:207–10.

10          Koh R, Pukallus M, Kularatna S, et al. Relative cost-effectiveness of home visits and telephone contacts in preventing early childhood caries. Community Dent Oral Epidemiol 2015;43:560–8. doi:10.1111/cdoe.12181

11          Pukallus M, Plonka K, Kularatna S, et al. Cost-effectiveness of a telephone-delivered education programme to prevent early childhood caries in a disadvantaged area: a cohort study. BMJ Open 2013;3(5):e002579. doi:10.1136/bmjopen-2013-002579

12          M S, R W, Eg K, et al. Cost-effectiveness of a disease management program for early childhood caries. J Public Health Dent 2014;75:24–33. doi:10.1111/jphd.12067

13          Mariño R, Fajardo J, Morgan M. Cost-effectiveness models for dental caries prevention programmes among Chilean schoolchildren. Community Dent Health 2012;29:302–8.

14          Duangthip D, Chen KJ, Gao SS, et al. Managing early childhood caries with atraumatic restorative treatment and topical silver and fluoride agents. Int J Environ Res Public Health 2017;14(10):1204. doi:10.3390/ijerph14101204

15          Smales RJ, Yip HK. The atraumatic restorative treatment (ART) approach for primary teeth: review of literature. Pediatr Dent 2000;22:294–8.

16          Wakshlak RB-K, Pedahzur R, Avnir D. Antibacterial activity of silver-killed bacteria: the “zombies” effect. Sci Rep 2015;5:9555. doi:10.1038/srep09555

17          Brostek AM, Walsh LJ. Minimal intervention dentistry in general practice. Oral Health Dent Manag 2014;13:285–94.

18          Slayton RL, Urquhart O, Araujo MWB, et al. Evidence-based clinical practice guideline on nonrestorative treatments for carious lesions: A report from the American Dental Association. J Am Dent Assoc 1939 2018;149:837-849. doi:10.1016/j.adaj.2018.07.002

19          Hendre AD, Taylor GW, Chávez EM, et al. A systematic review of silver diamine fluoride: Effectiveness and application in older adults. Gerodontology 2017;34:411–9. doi:10.1111/ger.12294

20          Monse B, Heinrich-Weltzien R, Mulder J, et al. Caries preventive efficacy of silver diammine fluoride (SDF) and ART sealants in a school-based daily fluoride toothbrushing program in the Philippines. BMC Oral Health 2012;12:52. doi:10.1186/1472-6831-12-52

21          Contreras V, Toro MJ, Elías-Boneta AR, et al. Effectiveness of silver diamine fluoride in caries prevention and arrest: a systematic literature review. Gen Dent 2017;65:22–9.

22          Yee R, Holmgren C, Mulder J, et al. Efficacy of silver diamine fluoride for arresting caries treatment. J Dent Res 2009;88:644–7. doi:10.1177/0022034509338671

23          Llodra JC, Rodriguez A, Ferrer B, et al. Efficacy of silver diamine fluoride for caries reduction in primary teeth and first permanent molars of schoolchildren: 36-month clinical trial. J Dent Res 2005;84:721–4. doi:10.1177/154405910508400807

24          Rosenblatt A, Stamford TCM, Niederman R. Silver diamine fluoride: a caries “silver-fluoride bullet.” J Dent Res 2009;88:116–25. doi:10.1177/0022034508329406

25          Crystal YO, Niederman R. Silver diamine fluoride treatment considerations in children’s caries management. Pediatr Dent 2016;38:466–71.

26          Mattos-Silveira J, Floriano I, Ferreira FR, et al. New proposal of silver diamine fluoride use in arresting approximal caries: study protocol for a randomized controlled trial. Trials 2014;15:448. doi:10.1186/1745-6215-15-448

27          Crystal YO, Marghalani AA, Ureles SD, et al. Use of silver diamine fluoride for dental caries management in children and adolescents, including those with special health care needs. Pediatr Dent. 2017; 39(5):135-145.

28          Duangthip D, Wong MCM, Chu CH, et al. Caries arrest by topical fluorides in preschool children: 30-month results. J Dent 2018;70:74–9. doi:10.1016/j.jdent.2017.12.013

29          Gao SS, Zhao IS, Hiraishi N, et al. Clinical trials of silver diamine fluoride in arresting caries among children: A Systematic Review. JDR Clin Transl Res 2016;1:201–10. doi:10.1177/2380084416661474

30          Fung MHT, Duangthip D, Wong MCM, et al. Randomized clinical trial of 12% and 38% silver diamine fluoride treatment. J Dent Res 2018;97:171–8. doi:10.1177/0022034517728496

31          Zhi QH, Lo ECM, Lin HC. Randomized clinical trial on effectiveness of silver diamine fluoride and glass ionomer in arresting dentine caries in preschool children. J Dent 2012;40:962–7. doi:10.1016/j.jdent.2012.08.002

32          Chibinski AC, Wambier LM, Feltrin J, et al. Silver diamine fluoride has efficacy in controlling caries progression in primary teeth: A systematic review and meta-analysis. Caries Res 2017;51:527–41. doi:10.1159/000478668

33          Chu CH, Lo ECM, Lin HC. Effectiveness of silver diamine fluoride and sodium fluoride varnish in arresting dentin caries in Chinese pre-school children. J Dent Res 2002;81:767–70. doi:10.1177/0810767

34          Narain JP. Health of tribal populations in India: How long can we afford to neglect? Indian J Med Res 2019;149:313–6. doi:10.4103/ijmr.IJMR_2079_18

35          Dikmen B. ICDAS II criteria (International Caries Detection and Assessment System). J Istanb Univ Fac Dent 2015;49:63–72. doi:10.17096/jiufd.38691

36          Tanzer J, Thompson A, Milgrom P, et al. Diammino Silver Fluoride Arrestment of Caries Associated with Anti-microbial Action. IADR/PER General Session (Barcelona, Spain) 2010.